# The Evolution of Cancer Immunotherapy

**DOI:** 10.3390/vaccines9060614

**Published:** 2021-06-08

**Authors:** Meshaal Khan, Ajay V. Maker, Shikha Jain

**Affiliations:** Oncology Department, University of Illinois at Chicago, Chicago, IL 60607, USA; amaker@uic.edu (A.V.M.); sjain03@gmail.com (S.J.)

**Keywords:** vaccines, immunotherapy, cancer, CAR-T, monoclonal antibodies, checkpoint inhibitor

## Abstract

Immunotherapy has changed the environment of cancer treatment by providing new and efficacious therapy options for many solid and hematologic malignancies. Although not a new field of oncology, immunotherapy has quickly developed into one of the most flourishing fields in medicine. In this review article, we explore key discoveries which helped to shape our current understanding of the immune system’s role in neoplasms. Many landmark developments include the advancements in checkpoint inhibitors, monoclonal antibodies, CAR-T cells and anti-cancer vaccines. We also explore the drawbacks and efficacy of various categories of immunotherapy. Ongoing investigations within immunotherapy, such as the gut microbiome, combining checkpoint inhibitors and gene sequencing, continue to personalize treatments for cancer patients, providing exciting and endless possibilities for the future.

## 1. Introduction

Over the past few decades, immunotherapy has been at the forefront of cutting-edge developments and discoveries for cancer treatments. New and promising treatments for malignancies with historically poor prognosis have been approved in an expedited manner compared to traditional therapies, and the landscape of cancer care is constantly and rapidly evolving. The evolution of immunotherapy for cancer care has been ongoing for centuries, with many key players and discoveries. There have been reports dating back to 1777 of attempts to inject various pathogens in order to elicit an immune response against neoplasia [1]. The first successful experiment was achieved by William Coley in 1891, who is often referred to as ‘the Father of Immunotherapy’. He injected streptococcal species into patients with metastatic soft tissue sarcomas and noted tumor regression [2]. He hypothesized that malignancy could be fought using a severe infection to activate the immune system. His injections are known today as ‘Coley’s toxins’ [3]. It was not until the 1950s that the concept of immunosurveillance was proposed. Burnet and Thomas were credited for the immunosurveillance hypothesis, in which the immune system actively works to recognize and eliminate neoplasia through tumor-associated antigen (TAA) recognition [4,5,6]. Currently, it is widely accepted that cancer cells are identified and eradicated by a functioning immune system, although these cells arise from a normal host cell rather than an exogenous insult [7]. Immunosurveillance of cancer can be summarized by the three Es: elimination, equilibrium and escape [8]. ‘Elimination’ refers to the recognition and destruction of tumor cells by innate and adaptive immunity. ‘Equilibrium’ describes the immune attack on tumor cells occurring simultaneously with tumor cells mutating and remodeling to resist attack. Lastly, mutated tumor cells ‘escape’ recognition by the immune system and continue to proliferate. This interaction between cancer cells and immunosurveillance remains crucial to the successes and failures of immunotherapeutic agents.

## 2. Cytokine Therapy

An earlier discovery in the field of cancer immunotherapy was the role of cytokines, which are small proteins utilized in cellular signaling [9]. In 1974, ‘T cell growth factor’, or IL-2, was discovered [10]. IL-2 is produced by several cells including CD4+ and CD8+ T cells, and is crucial in T cell differentiation and growth, immune memory and maintaining regulatory T cells to prevent autoimmunity [11]. IL-2 was later cloned in 1983 and was studied in mice models, noting metastatic malignancy regression in syngeneic sarcoma and melanoma [12,13,14]. This ultimately led to FDA approval in 1992 for the treatment of metastatic renal cell carcinoma, making IL-2 the first cancer immunotherapy in humans [15]. IL-2 is associated with significant toxicities, including capillary leak syndrome and multiple organ dysfunction, limiting its use to specialized centers [16].

## 3. Checkpoint Inhibition

The knowledge of innate tumor suppression by a functioning immune system has paved the way for one of medicine’s most profound discoveries: checkpoint inhibition. Through complex mechanisms, blocking the immune system’s regulatory checks and balances using selective antibodies can provide antitumor activity with more tolerable toxicities [9].

Shortly after IL-2 approval, checkpoint inhibitors moved to the forefront of cancer research. The discovery of cytotoxic T lymphocyte-associated antigen 4, or CTLA-4, paved the way for all future checkpoint inhibitors. CTLA-4 is expressed on T cells and assists in controlling immune hyperactivation and host damage [17]. Briefly, CTLA4 acts in a complex fashion with CD28 and B7; CTLA-4 is a homolog of CD28 on T cell receptors and competitively binds to antigen presenting cells [18]. It also does not cause a stimulatory signal, thus blocking downstream interactions of TCR with APCs and ultimately deactivating T cells [19].

This checkpoint, discovered by James Allison, led to the approval of ipilimumab for metastatic melanoma in 2011 as the first checkpoint inhibitor for cancer treatment. Ipilumumab is an anti-CTLA-4 monoclonal antibody and works by directly blocking CTLA-4, making way for downstream T cell activation, proliferation and eventual tumor destruction [20,21,22]. Following CTLA-4’s discovery, Ishida et al. uncovered both programmed death-1 and programmed death ligand-1 (PD-1/PDL-1) in 1992 [23]. T cell lymphocytes express PD-1 and, when bound to PDL-1, act to hinder T cell function and prevent auto-destruction of the tissue. Further investigations revealed an overexpression of PD-1/PDL-1 in certain tumor cells, thus avoiding this particular pathway of immune-mediated tumor destruction [24]. This led to the discovery of pembrolizumab (Keytruda), an anti-PD-1 antibody intended for metastatic or unresectable melanoma previously treated with ipilimumab, or BRAF V600 mutated patients previously treated with a BRAF inhibitor [25]. Figure 1 demonstrates the mechanisms of PD-1 and CTLA-4 on T-cells and tumor cells.

Currently, there are multiple approved PD-1 inhibitors, including pembrolizumab (Keytruda), nivolumab (Opdivo) and cemiplimab (Libtayo), as well as PDL-1 inhibitors atezolizumab (Tecentriq), avelumab (Bavencio) and durvalumab (Imfinzi). Despite many cancers expressing high levels of PD-1, several mechanisms can limit the cancer’s response to checkpoint inhibitors. A phenomenon known as ‘self-neutralization’ can occur, whereby cancer cells concomitantly express both PD-1 and its ligand, PDL-1. This results in the binding of PD-1/PDL-1, thus eliminating the drug’s primary target [26]. Combination checkpoint inhibitor therapy with CTLA-4 and PD-1 blockade proposes a concomitant tumor attack through two distinct mechanisms. This suggests that more patients respond to the treatment overall, in addition to a more durable response. A combination of ipilimumab and nivolumab is currently FDA-approved for several malignancies, including first-line therapy for NSCLC with PDL-1 expression >1%. However, increased toxicities remain an obstacle for many combination regimens [27]. There are over 2000 ongoing trials for PD-1/PDL-1 inhibitors in various malignancies and combinations, highlighting the dramatic focus on this groundbreaking treatment [28].

## 4. Antitumor Monoclonal Antibodies

The concept that cancer cells express certain potentially targetable antigens has paved the way for humanized antibodies as a treatment strategy for malignancy. Antibodies can target and destroy specific tumor antigens by interacting with immune cells through the antibody’s fragment crystallizable (Fc) region. The Fc region on an antibody can bind to various Fc receptors (FcRs) that are found on natural killer cells, neutrophils, eosinophils, dendritic cells, or monocytes. Depending on the antibody class, this immune cell–antibody interaction can lead to tumor cell death in several ways, including complement-dependent cytotoxicity (CDC), antibody-dependent cellular cytotoxicity (ADCC) and antibody-dependent cellular phagocytosis (ADCP) [29]. Kohler and Milstein won the Nobel Prize in Physiology or Medicine in 1984 for their hybridoma technique [30]. This breakthrough has led to the development of many antitumor monoclonal antibodies (MABs), greatly influencing cancer treatment in the previous decades. The first therapeutic antibody in oncology is still widely used today: rituximab. The anti-CD20 MAB was approved in 1997 and was initially FDA-approved for CD20+ Non-Hodgkin’s lymphoma [31]. CD20, the target of rituximab, is a transmembrane protein expressed early in B cell development and hypothesized to aid in B cell activation while generally not being expressed by other cells. Although approved over 20 years ago, rituximab is still used to treat the vast majority of B cell NHL and is now approved for chronic lymphocytic leukemia, rheumatoid arthritis, and certain autoimmune conditions such as pemphigus vulgaris. While very tolerable, rituximab does have side effects, such as flu-like symptoms, infusion reactions, mucocutaneous reactions and a reactivation of the hepatitis B virus.

Another groundbreaking development in cancer therapy was the development and approval of trastuzumab (Herceptin), which is a monoclonal antibody that targets human epidermal growth factor receptor 2 (HER2) [32]. HER2 is a proto-oncogene and is classified as a transmembrane receptor tyrosine kinase. It is structurally similar to EGFR, with many downstream signaling effects. Slamon et al. discovered that increased expression of tumor-associated HER2 was associated with a shortened survival in patients with ovarian and breast cancer [33]. HER2 overexpression is found in up to 30% of invasive breast cancer cases and is a poor prognostic indicator, both for overall survival and the time taken to relapse [34]. Trastuzumab binds to extracellular segment IV of HER2 and works in multiple ways, including HER2 degradation, antibody-dependent cellular cytotoxicity and cell cycle arrest through inhibition of MAPK and PI3K/Akt pathways [35]. Similar to other monoclonal antibodies, trastuzumab can cause infusion reactions and pulmonary toxicity. Trastuzumab can also cause significant cardiomyopathy, which is speculated to be due to HER2 involvement in cardiomyocyte cell regulation and signaling. In contrast to anthracycline-induced cardiomyopathy, this cardiomyopathy is often reversible with the discontinuation of the medication. Since approval in 1998 for HER2 overexpressing breast cancer and HER2 overexpressing metastatic gastric or GE junction adenocarcinoma, trastuzumab has provided both substantial progression-free and overall survival benefits for patients with HER2 overexpressing breast cancer and is a fundamental part of anti-HER2 treatment [36].

There are currently over 500 approved and investigational MABs engineered against solid and hematologic malignancies, as well as benign hematologic conditions, autoimmune and chronic diseases. Table 1 illustrates several agents routinely used in practice for various solid and hematologic malignancies. While MABs are a mainstay in current cancer treatment, challenges remain in delivering effective therapies. Due to the dynamic nature of cancer cells and their ongoing mutations, any acquired resistance against MABs results in treatment failure [37]. Strategies being implemented include combination drug therapy with cytotoxic or other immunotherapy agents, or the advent of antibody–drug conjugates such as Kadcyla. Kadcyla, or ado-trastuzumab emtansine (T-DM1), specifically includes trastuzumab linked with emtansine, which is an anti-microtubule cytotoxic chemotherapy drug. This antibody–drug conjugate works by degrading after it is internalized into the cell following HER2 receptor binding, causing the release of DM-1 and thus further cytotoxic cell death [38]. Kadcyla is currently approved for metastatic HER2+ breast cancer following treatment with trastuzumab and taxanes, as well as for early HER2+ breast cancer in the adjuvant setting following treatment with trastuzumab. Antibody–drug conjugates such as Kadcyla are providing promising options for many malignancies. Monoclonal antibodies remain one of the most exciting and evolving areas of cancer-directed therapy.

## 5. CAR-T Cell Therapy

Chimeric antigen receptor T (CAR-T) cell therapy has been at the forefront of the novel treatment of hematologic malignancies over the past few decades. In the early 1990s, Eshhar et al. sought to bypass the limitations of T cell silencing caused by the tumor microenvironment [39]. In doing so, they developed the first chimeric antigen receptor, paving the way for an astounding new cancer treatment. In summary, a patient’s T cells are collected, and, using a modified inactive virus, the T cells are essentially reprogrammed to produce ‘special’ receptors called chimeric antigen receptors, or CARs, on their surface [40]. These reprogrammed chimeric T cells are then infused back into the patient. The CARs then redirect T cell function and specifically allow the engineered T cells to latch onto cancer cells, propagating and facilitating their death. Although this therapy has been researched for decades, the first CD-19-directed CAR-T cells (Kymriah) were approved relatively recently, in 2017, for relapsed, refractory, acute lymphoblastic leukemia [41]. Currently, CAR-T cells are FDA-approved for B cell lymphomas and ALL. CAR-T cells are revolutionizing the treatment of hematologic malignancies as they have shown remarkable response rates up to 94% [42].

Thus far, FDA-approved CAR-T cell therapy has been limited to B cell malignancies expressing CD19. This has limited the successful development of CARs for other hematologic and solid tumors. A key player in this limitation is tumor-associated antigen heterogeneity of solid tumor cells, making it difficult to engineer a CAR that is successful against all malignant cells of a particular cancer [43]. Another difficulty with solid tumors is the ability of the CAR-T cell to infiltrate the blood, pass through the vasculature and ultimately reach the target tumor. Ligand-11 and 12 chemokines are important for this CAR-T cell movement and are under-expressed in many solid tumors [43]. This suggests the possible need for regional administration of CAR-T therapy, such as in the breast, pleura or brain. Despite these microenvironment limitations, current studies are investigating a multitude of potential antigens for solid tumors, such as HER2 for breast cancer and prostate-specific membrane antigens (PMSAs) in prostate cancer [44,45]. Although they have high potential, CAR-T cells can harbor significant toxicity, including severe cytokine release syndrome (CRS), as well as substantial neurotoxicity. This has limited the widespread use of CAR-T cell regimens and currently restricts therapy to high-volume centers trained in managing toxicities [42]. Nevertheless, the concept of engineering a CAR-T cell against any antigen presents endless possibilities for further treatment in both solid and hematologic malignancies.

### Bispecific Antibodies/BiTEs

While monoclonal antibodies have become a backbone of cancer treatment, bispecific antibodies are allowing for more robust immunogenicity as well as a more targeted anti-cancer response. Bispecific antibodies, or BsAbs, have two separate antigen binding sites. In immuno-oncology specifically, one binding site is directed to promote T cells while the other is attached to a tumor-specific antigen [46]. In the US, blinatumomab was approved in 2014 for Ph-negative relapsed or refractory B cell acute lymphoblastic leukemia. Blinatumomab harbors two binding sites per molecule: one for CD3 (T cells) and one for CD19 (B cells). Currently, it remains the only FDA-approved bispecific antibody. However, dozens are being investigated in clinical trials. Blinatumomab is also considered a bispecific T cell engager, or BiTE. These molecules are a class of bispecific antibodies designed to attach to both cytotoxic T cells and cancer cells by expressing two linked single-chain variable fragments [47]. One scFv binds to CD3, and the other to a tumor-specific antigen, as previously discussed with the structure of blinatumomab. This allows for increased cytotoxic T cell eradication of neoplastic cells.

Both bispecific antibodies and CAR-T cells are used to harness T-cell-directed immunotherapy, thus fueling a comparison of both methods. While CAR-T cells are genetically engineered for a specific target and patient, with excellent resulting responses against hematologic malignancies, they have limitations, including cost, time and the expertise required in creating and delivering CAR-T therapy. Few centers are participating in CAR-T therapy due to both its extreme cost and the additional training required, thus limiting its access largely to tertiary centers in major metropolitan cities. Bispecific antibodies and BiTEs are available ‘off the shelf’, thus lowering costs and increasing access for many patients when compared with CAR-T. The predetermined antigen, however, limits the specificity and personalization of bispecific antibodies. Both CAR-T and BiTE cells have side effects that include cytokine release syndrome and neurotoxicity, creating a need for specialized training in managing these medications.

## 6. Vaccines

As further knowledge regarding host immunity against tumor antigens emerges, vaccine-induced immunotherapy is, in theory, an ideal treatment. Various vaccines targeting tumor-associated antigens (TAAs) have been developed in vitro, ranging from synthetic peptide vaccines, viral-based, RNA/DNA vaccines and cell-based vaccines.

### 6.1. Preventative Vaccines

Preventing tumor cells from growing and proliferating could be an ideal solution to our long battle with cancer, and this has been studied for the past few decades. However, the development of preventive cancer vaccines has also been plagued by a lack of tumor antigen specificity and the similarity of tumor cells to self cells, which could lead to overwhelming autoimmune phenomena. Nevertheless, there are currently two approved cancer prevention vaccines: the human papillomavirus (HPV) vaccine and the hepatitis B virus (HBV) vaccine [47]. Both of these vaccines target viruses with oncogenic potential HPV16 and 18 give rise to 70% of cervical cancer cases. Although the exact mechanism of tumorigenesis is unknown, chronic hepatitis B infection is directly linked to hepatocellular carcinoma and cirrhosis. Importantly, these preventative measures have remarkably lowered infection rates and cancer diagnoses worldwide.

### 6.2. Therapeutic Vaccines

The utilization of vaccines for treatment rather than prevention is rather unique to the field of oncology. Bacillus Calmette–Guerin (BCG) is widely used for the treatment of non-muscle invasive bladder cancer (NMIBC) and has been for four decades [47]. BCG is a live attenuated vaccine derived from Mycobacterium Bovis. It was initially developed as a vaccine against tuberculosis, but was later tested broadly as a treatment against various types of cancer [48,49]. It was approved in the 1970s as an intravesicular treatment for NMIBC and is still considered to be the standard therapy for preventing the progression or recurrence of high-risk disease [49]. Despite this promising treatment, there have been few successful antitumor vaccines used as monotherapy. One reason is due to the lack of specificity of tumor antigens, as many are expressed on cancer cells as well as healthy tissue, posing the risk for significant autoimmunity and organ damage [48]. Another barrier for therapeutic vaccines is the downregulation of the immune system caused by cancer cells and the altered self-antigens that the tumors create. There is a certain balancing act necessary for a vaccine to be highly toxic to cancer cells while preserving and protecting normal tissue cells.

## 7. Oncolytic Viruses

A fairly novel class of immunotherapy includes using viruses to attack and infiltrate malignant cells, either through lab modification or through pre-existing means. These oncolytic viruses work through the direct lysis of tumor cells as well as the activation of innate and adaptive immune mechanisms. Prior approvals include Rigvir (2004, Latvia) and Oncorine (2005, China); however, concerningly, these were met with adverse effects in healthy tissue [50]. In 2015, Talimogene laherparepvec (TVEC) was the first approved oncolytic virus in the United States [51]. TVEC consists of a herpes simplex virus-1 encoded with two gene copies, GM-CSF, promoting dendritic cell activation. Mechanistically, TVEC suppresses the protein kinase-R (PKR) pathway and works to upregulate type-1 IFN. This allows the virus to directly attack cancer cells and induce immunogenicity. Currently, TVEC is approved for advanced melanoma (stage IIIB-IV) based on a randomized controlled phase III trial which showed an improved ORR and a durable response [52].

Current FDA-approved therapeutic cancer vaccines include Sipuleucel-T, a dendritic cell-based vaccine, used for minimally symptomatic metastatic castrate-resistant prostate cancer, and T-VEC, an intralesional HSV-1-derived oncolytic viral vaccine intended for unresectable, recurrent melanoma [53]. Although the phase III trial of T-VEC demonstrated an overall survival benefit, this was not statistically significant (*p* = 0.051). The IMPACT trial demonstrated a 4-month median survival improvement in the Sip-T group compared to the placebo, and the overall survival was statistically significant (*p* = 0.032) [54].

### 7.1. Neoantigen Vaccines

Current research in immunotherapeutic vaccines includes neoantigen-based personalized vaccines. Personalized neoantigens arise from mutations of cancer cells exclusively, providing new epitopes that are considered non-self and can ultimately be recognized by immunosurveillance [53]. In theory, these neoantigens are exclusively expressed by tumor cells and are highly immunogenic, making them a perfect target for a directed vaccine. Clinical trials are underway to identify neoantigens through exome sequencing, to create a personalized vaccine, and to ultimately assess safety and efficacy.

### 7.2. Combination Vaccine Therapies

Combining checkpoint inhibitors with vaccines may prove synergistic and more efficacious than vaccine monotherapy. Various trials have evaluated Talimogene laherparepvec (T-VEC) with checkpoint inhibitors. The mechanism of T-VECs includes the lysis of tumor cells after entry and the release of granulocyte–macrophage colony stimulating factor (GM-CSF) and tumor-derived antigens, enhancing the T cell response on the tumor [54]. Mechanistically, an anti-cancer vaccine combined with a checkpoint inhibitor would provide a dual attack on cancer cells with optimum T cell recruitment and less T cell exhaustion. A 2016 phase II clinical trial enrolled 198 patients with unresectable melanoma to either T-VEC plus ipilumumab or ipilumumab monotherapy [55]. This randomized trial met its primary endpoint with an increased ORR in the TVEC plus ipilumumab group vs. ipilumumab alone. Additionally, the adverse effect profile mimicked that of ipilumumab monotherapy, suggesting no additional toxicity by adding T-VEC. A similar study of 21 patients evaluated T-VEC plus pembrolizumab for unresectable stage IIIB to IV melanoma [56]. The study confirmed an ORR of 62%, without additional toxicity, when compared to pembrolizumab alone. Currently, there is a multicenter phase III trial nearing completion that is evaluating pembrolizumab administration with T-VEC in unresectable melanoma (KEYNOTE-034).

### 7.3. Future Studies in Vaccines

As the understanding of vaccine techniques and modalities increases, so does the opportunity for expanding vaccines as cancer therapies. In the field of neuro-oncology, glioblastoma multiforme (GBM) and high-grade gliomas have poor prognoses overall. In 2019, initial phase II data of a personalized cancer vaccine called AV-GBM-1 were published, showing an increased PFS when used as an adjunct to surgery and concurrent chemoradiation [57]. AV-GBM-1 is a novel personalized cancer vaccine consisting of autologous dendritic cells loaded with host tumor-associated antigens. The trial has a tentative completion date of 2023. With the recent approval of mRNA vaccines for SARS-CoV-2, research is underway for the use of mRNA vaccines in malignancy, with several in preclinical or phase I development.

## 8. Future/In Process

The rapidly changing landscape of cancer treatments highlights the impact that immunotherapy has made, as well as the hope for better responses to treatment with more tolerable side effects. Current endeavors include reinvestigating known treatments in novel ways, such as combining checkpoint inhibitors with therapeutic cancer vaccines to assess response [58]. Multiple phase I studies in various solid and liquid tumor types are underway. In the age of personalized medicine, further efforts are underway to personalize cancer treatment. The gut microbiome could provide vital information regarding response to therapy, as well as potential side effects [59]. An ongoing observational study at Cambridge will attempt to correlate an individual’s microbiome with the efficacy and toxicity of various immunotherapies [60]. The ability to predict one’s response to immunotherapy remains perplexing, and biomarkers remain non-uniform amongst cancer types. For example, PD-L1 testing and expression is used to gauge response in lung cancer. However, it does not gauge response in other malignancies such as melanoma [61]. The importance lies in the fact that 44% of patients in the United States with cancer are eligible for checkpoint inhibition, versus 13% estimated to have a response, making it of the utmost importance to identify this population [62]. Additional combinations of cytotoxic chemotherapy with immunotherapy, as well as combining various immunotherapy agents, are constantly being investigated. Adoptive cell therapy, including CAR-T treatment, is currently being investigated in solid tumors. However, this has proven to be challenging thus far due to the specific features of solid tumor microenvironments [63]. Combining checkpoint inhibition, such as a PD-1/PDL-1 blockade, with CAR-T therapy, may prove beneficial and is also under investigation [64]. Given the wide array of known immunotherapy agents, there continue to be different efforts to combine these agents and ultimately increase cytotoxic response.

## 9. Conclusions

In many ways, immunotherapy has forever changed the way we approach the practice of hematology and oncology and has emerged as a powerful tool in cancer care. The possibilities for further innovation involving these novel therapeutic agents will continue to shape the way we personalize cancer care, as more is discovered about how these agents both work and enhance traditional therapies, as well as how they provide new ways to approach malignancies that were previously difficult to treat. Through years of research and innovation, immunotherapy now has many sub-categories, including, but not limited to, checkpoint inhibition, monoclonal antibodies, CAR-T cells and oncolytic viruses. Further exploratory research includes: immunotherapy in the neoadjuvant and adjuvant setting; combination therapy of different types of immunotherapeutic agents with various mechanisms of action; the utility of immunotherapy in combination with other targeted therapies such as radiation; combination therapy with immunotherapy and more traditional therapeutic agents. This research leaves the space wide open for innovation, with countless opportunities to both potentially incorporate these regimens into various permutations of standard of care and to enhance current practices. Further study on which patients will most benefit from certain agents, as well as the optimal sequencing of agents, will better allow the personalization of cancer care as more is understood on these therapeutic interventions. As the indications for currently approved agents continue to expand, and new agents are introduced and studied, the field of cancer immunotherapy continues to grow. These agents have changed the playing field, and as our knowledge of the pathophysiology and mechanism of action and interaction continues to evolve, so do the possibilities for the utilization and implementation of these novel agents. The progress made in oncology care as a direct result of the development and implementation of these immunotherapeutic agents has produced a watershed moment for cancer care that truly represents the power of studies that are founded in science and focused on clinical practice and outcomes. The evolution of immunotherapy is an optimal example of the power of the ‘bench to bedside’ approach that forms the foundation of hematology and oncology. Given the advancements made in this space, future possibilities remain endless.

## Figures and Tables

**Figure 1 vaccines-09-00614-f001:**
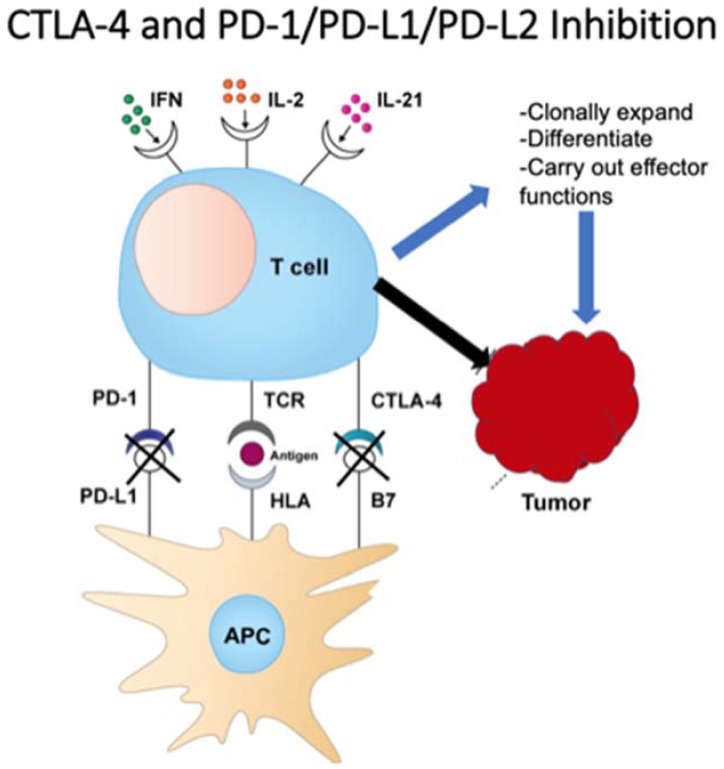
T cell function on tumor cells.

**Table 1 vaccines-09-00614-t001:** Commonly used MABs.

Drug Name	Target	Malignancy Approved (Single Agent or in Combination)	Approval Year
Atezolizumab	PD-L1	Multiple solid tumors	2016
Bevacizumab	VEGF-A	Multiple solid tumors	2004
Brentuximab vedotin	CD30	Hodgkin’s lymphoma; Anaplastic LCL; PTCL	2011
Blinatumomab	CD3, CD19	Acute lymphoblastic leukemia	2014
Cemiplimab	PD-1	Cutaneous squamous cell carcinoma	2018
Cetuximab	EGFR	Colorectal cancer (K-RAS wildtype); NSCLC, SCC head and neck	2004
Daratumumab	CD38	Multiple myeloma	2015
Durvalumab	PD-L1	Urothelial carcinoma, NSCLC, small cell lung cancer	2017
Elotuzumab	SLAMF7	Multiple myeloma	2015
Gemtuzumab ozogamicin	CD33	Acute myeloid leukemia	2000
Ipilimumab	CTLA-4	Multiple solid tumors	2011
Isatuximab	CD38	Multiple myeloma	2020
Mogamilizumab	CCR4	Mycosis fungoides or Sezary syndrome, CTCL, T cell leukemia/lymphoma	2018
Nivolumab	PD-1	Multiple solid tumors	2014
Obinutuzumab	CD20	Chronic lymphocytic leukemia, follicular lymphoma	2013
Panitumumab	EGFR	Colorectal cancer	2006
Pembrolizumab	PD-1	Multiple solid tumors	2014
Pertuzumab	HER2	Breast cancer (HER2+)	2012
Ramucirumab	VEGFR2	Multiple solid tumors	2014
Rituximab	CD20	Multiple hematologic malignancies and autoimmune diseases	1997
Trastuzumab	HER2	Breast cancer (HER2+), gastric/GEJ adenocarcinoma (HER2+)	1998

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
