# Peer review of "The Evolution of Cancer Immunotherapy"

_vaccines, 2021, doi:10.3390/vaccines9060614_

Round 1
Reviewer 1 Report
This is a well written review of a potentially interesting topic, and therefore the target audience is relatively broad. On the positive side, there may be interest in the likely significant biological relevance for the treatment of cancers. As described in the specific comments below, I think there are a few aspects of this topic that should be discussed in greater depth to give the reader a better idea of the complexities of this topic. On the negative side, while the topic is interesting, the molecular mechanisms of immunotherapy is ill-defined for check point inhibitors, CAR-T cells and bi-specific antibodies. Despite these limitations, the review may be useful for larger audiences.
Specific Comments:
It is noteworthy that the delivery and effectiveness of immunotherapy are highly dependent on the cancer type, grade, predictive response rate, and expression of critical biomarkers; however, patient response rates can still vary.
- There are clinical findings that even though PD-1 is highly expressed in tumor cells in some patients, the effect of PD-1/PD-L1 immune checkpoints inhibitors is still poor. Studies have shown that this situation occurs because these tumor cells express both PD-L1 and PD-1. Before PD-1/PD-L1 immune checkpoints inhibitors act on PD-L1 of tumor cells, PD-L1 has been bound to its own PD-1, resulting in the loss of the target of PD-1/PD-L1 immune checkpoints inhibitors. This mechanism is called the PD-1 and PD-L1″self-neutralization" of immune checkpoints inhibitors pathways remains a challenge which authors did not address in the manuscript.
- CAR T cells-based therapies is improving outcomes for many patients with advanced malignancies, but success has thus far been limited to B cell malignancies. Extending this approach to other cancers will require the understanding of tumor heterogeneity and the tumor microenvironment. I would urge authors elaborate this with literature.
Reviewer 2 Report
Khan, Maker and Jain's article is a review on cancer immunotherapy. It summarizes advances in various subfields such as monoclonal antibodies, vaccines, CAR-T cells and oncolytic viruses and other antibody-based immunomodulators.
- Although scientifically correct, this article is sorely lacking in recent data. Indeed, only 5 articles from 2020 are cited for 57 total references. In view of the rapid advances in the field, especially with hundreds of clinical trials in progress, the reader might expect to receive a more recent update. What is more, the latest research in preclinical development would have added value to the article. Finally, the table on page 5 which lists the monoclonal antibodies is very incomplete, because it lacks among others Isatuximab, Ofatumumab, Olaratumab, Necitumumab etc.
- The arrangement of the different therapeutic modalities is not understandable, so why put in the same paragraph the cytokines with the monoclonal antibodies directed against the immune checkpoint inhibitors, or the CAR-T cells with the BiTES?
- There is also a lack of discussion around the pathways induced by Fcgamma-Rs, including ADCC and ADCP, which have been shown to be important in the mode of action of monoclonal antibodies.
- Most important to me is the too short and too vague discussion of anti-tumor vaccination, more so for an article submitted to a journal called vaccines. An in-depth discussion centered on therapeutic combinations and vaccination would have been appreciated.
- The title of the Figure contains a typing error
Author Response
Although scientifically correct, this article is sorely lacking in recent data. Indeed, only 5 articles from 2020 are cited for 57 total references. In view of the rapid advances in the field, especially with hundreds of clinical trials in progress, the reader might expect to receive a more recent update. What is more, the latest research in preclinical development would have added value to the article. Finally, the table on page 5 which lists the monoclonal antibodies is very incomplete, because it lacks among others Isatuximab, Ofatumumab, Olaratumab, Necitumumab etc.
Thank you for the constructive feedback on our review. Although this review was focused on the 'evolution' of immunotherapy and required many sources prior to 2020, we agree more recent data was lacking in our review. We have added research in preclinical development including AV-GBM-1 in glioblastoma multiforme and a brief discussion on mRNA vaccines. The table was re-named 'commonly used MABs' to reflect monoclonal antibodies commonly used in clinical practice rather than a complete list. We have added gemtuzumab and isatuximab to reflect this.
The arrangement of the different therapeutic modalities is not understandable, so why put in the same paragraph the cytokines with the monoclonal antibodies directed against the immune checkpoint inhibitors, or the CAR-T cells with the BiTES?
We have re-arranged the review to have separate paragraphs discussing cytokine therapy, checkpoint inhibitors, monoclonal antibodies, CAR-T cells and BiTES.
There is also a lack of discussion around the pathways induced by Fcgamma-Rs, including ADCC and ADCP, which have been shown to be important in the mode of action of monoclonal antibodies.
Thank you for noticing this oversight. We have added a brief discussion on this in paragraph one of 'anti tumor monoclonal antibodies'.
Most important to me is the too short and too vague discussion of anti-tumor vaccination, more so for an article submitted to a journal called vaccines. An in-depth discussion centered on therapeutic combinations and vaccination would have been appreciated
We appreciate this feedback and agree with you. We have separated the vaccine section into further sub-sections to allow for a more extensive discussion regarding different types of vaccine therapy. We have added discussion on currently used and approved anti-cancer vaccines including TVEC and Sip-T and their mechanisms. We have added a discussion on combination vaccine therapies including previously studied combinations and pembrolizumab + TVEC which is currently undergoing phase III trial completion. We have also added a 'future studied in vaccines' section to discuss AV-GBM-1 and briefly discuss mRNA vaccines.
Round 2
Reviewer 1 Report
The authors addressed my comments and I would recommend the current version of manuscript for publication in Vaccines.
Author Response
Thank you for your feedback
Reviewer 2 Report
-there is a typo in the title of the figure 2
-please check if figures 1 and 2 should be so similar, in particular the right part of the figures are exactly the same
Author Response
Thank you for noticing this. We have omitted one of the figures (as they were quit similar) and fixed the typing error.